# Characterizing Glomerular Barrier Dysfunction with Patient-Derived Serum in Glomerulus-on-a-Chip Models: Unveiling New Insights into Glomerulonephritis

**DOI:** 10.3390/ijms25105121

**Published:** 2024-05-08

**Authors:** Shin Young Kim, Yun Yeong Choi, Eun Jeong Kwon, Seungwan Seo, Wan Young Kim, Sung Hyuk Park, Seokwoo Park, Ho Jun Chin, Ki Young Na, Sejoong Kim

**Affiliations:** 1Department of Internal Medicine, Seoul National University Bundang Hospital, Seongnam-si 13620, Republic of Korea; chlorella@hanmail.net (S.Y.K.); r2759@snubh.org (Y.Y.C.); dhkdud@hanmail.net (W.Y.K.); qq2ww3@naver.com (S.H.P.); 2Department of Internal Medicine, Seoul National University College of Medicine, Seoul 03080, Republic of Korea; dolce102003@gmail.com (E.J.K.); no1seokwoo@snubh.org (S.P.); kyna@snubh.org (K.Y.N.); 3Osong Medical Innovation Foundation, Cheongju-si 28161, Republic of Korea; seungwan@kbiohealth.kr (S.S.); mednep@snubh.org (H.J.C.)

**Keywords:** glomerulonephritis, kidney glomerulus, glomerulus-on-a-chip, glomerular filtration barrier, podocytes, glomerular disease models

## Abstract

Glomerulonephritis (GN) is characterized by podocyte injury or glomerular filtration dysfunction, which results in proteinuria and eventual loss of kidney function. Progress in studying the mechanism of GN, and developing an effective therapy, has been limited by the absence of suitable in vitro models that can closely recapitulate human physiological responses. We developed a microfluidic glomerulus-on-a-chip device that can recapitulate the physiological environment to construct a functional filtration barrier, with which we investigated biological changes in podocytes and dynamic alterations in the permeability of the glomerular filtration barrier (GFB) on a chip. We also evaluated the potential of GN-mimicking devices as a model for predicting responses to human GN. Glomerular endothelial cells and podocytes successfully formed intact monolayers on opposite sides of the membrane in our chip device. Permselectivity analysis confirmed that the chip was constituted by a functional GFB that could accurately perform differential clearance of albumin and dextran. Reduction in cell viability resulting from damage was observed in all serum-induced GN models. The expression of podocyte-specific marker WT1 was also decreased. Albumin permeability was increased in most models of serum-induced IgA nephropathy (IgAN) and membranous nephropathy (MN). However, sera from patients with minimal change disease (MCD) or lupus nephritis (LN) did not induce a loss of permeability. This glomerulus-on-a-chip system may provide a platform of glomerular cell culture for in vitro GFB in formation of a functional three-dimensional glomerular structure. Establishing a disease model of GN on a chip could accelerate our understanding of pathophysiological mechanisms of glomerulopathy.

## 1. Introduction

Glomerulonephritis (GN) is a complex kidney disease with varied etiological, pathological, and clinical symptoms. It is characterized by an impairment of the glomerular filtration barrier (GFB), which results in proteinuria, leading to end-stage renal failure and poor clinical outcomes, including renal death and increased overall mortality. Although immunoglobulin A nephropathy (IgAN), membranous nephropathy (MN), minimal change disease (MCD), and lupus nephritis (LN) have different etiologies, they are the most common forms of GN [1]. Unfortunately, there is still no specific cure available for most cases of GN.

Glomeruli, the blood-filtering capillary networks in the kidneys, are structured to retain circulating cells and macromolecules (such as albumin) and to remove metabolic wastes from the blood into an almost-protein-free urinary filtrate [2]. The filtration capacity of glomeruli is mediated by the GFB, an intricate trilayer filter comprising a glomerular basement membrane (GBM) sandwiched between endothelial cells and podocytes (the cells that regulate the selective permeability of the GFB) [3]. Among these components, podocytes are critical for the maintenance of glomerular morphology and function. Podocyte injury can lead to the occurrence of proteinuria, podocyte foot process effacement- and depletion, detachment from the GBM, glomerular filtration dysfunction, and ultimately loss of renal function, which is believed to be key to the pathogenesis of GN [4,5,6].

Although animal models are widely used to study human kidney biology for preclinical drug development and disease models of glomerulopathy, such models often yield results that are not directly applicable to humans due to species-specific differences in biochemical, physiological, developmental, and anatomical characteristics. Furthermore, animal experiments have issues such as multi-organ effect, individual differences, unstable reproducibility, poor reliability, low throughput, and increasing tendency for social exclusion [7,8]. Additionally, heterogeneity in disease phenotypes and varied response to drugs in different patients emphasize the need for more personalized disease modeling approaches. Thus, there is a critical need for alternatives to animal models that can closely replicate human physiology to predict the efficacy, safety, bioavailability, and toxicity of candidate therapeutics of glomerulopathy.

Recent advances in kidney-on-a-chip or renal microfluidic device technology have demonstrated great potential for kidney disease modeling and assessment of drug-induced kidney injury by recapitulating in vivo tissue characteristics. A glomerulus-on-a-chip research study reported that a unique culture environment mimicking the in vivo physiological environment can elicit more potential in cultured cells than a conventional static culture [9,10,11,12,13]. Thus, it is a promising tool as an alternative to animal experiments. In addition, the effort to develop in vitro glomerular models would greatly help advance our understanding of the mechanisms that underlie kidney development and facilitate the establishment of disease models to guide therapeutic discovery. Recently, cytoskeletal rearrangement and junctional injury in both podocytes and endothelium due to hypertension in a glomerular chip have been demonstrated [14]. A previous publication has described a glomerular chip model constituted by human amniotic fluid-derived podocytes and human glomerular endothelial cells using a three-lane organoplate [15]. Although this chip contains advanced cellular and protein GBM components and shows the glomerular barrier function, amniotic fluid kidney progenitor-derived podocytes are not routinely applicable for high-throughput use. Moreover, this set-up precludes stimulation or treatment of one specific cell type, as well as easy experimental separation and subsequent individual analysis of two cell types. Additionally, limited supply and invasive sourcing of cells along with ethical concerns limit the potential use of glomerulus-on-a-chip to mimic in vivo tissue development, physiology, and function.

To address these issues and further advance tissue-on-a-chip technologies and kidney disease outcomes, we developed a glomerulus-on-a-chip constructed by co-culturing human podocytes and glomerular endothelial cells in a novel compartmentalized three-dimensional (3D) microfluidic device. We also investigated podocyte injury induced by serum from a patient with GN on a chip to assess the capable of human glomerulus recapitulating device as a disease model for predicting responses to human GN.

## 2. Results

### 2.1. Clinical and Pathological Characteristics of Enrolled Patients

All patients were Asians clearly diagnosed with GN by renal biopsy and clinical manifestations. Characteristics of patients are presented in Appendix A. The UPCR (urine protein/creatinine ratio) levels of 4 out of 5 IgA patients were >500 mg/g, and for 1 patient (IgAN3), the level was 407.69 mg/g. IgAN2 and IgAN5 patients had low eGFR-MDRD (estimated glomerular filtration rate–modification of diet in renal disease) values of 6.6 mL/min/1.73 m^2^ and 20.1 mL/min/1.73 m^2^, respectively, and they underwent kidney transplantation. Two patients (IgAN1 and IgAN3) had an eGFR-MDRD of 60–89 mL/min/1.73 m^2^, and the IgAN4 patient had an eGFR-MDRD of 48 mL/min/1.73 m^2^. All patients with MN had high levels of UPCR > 500 mg/g. A total of 2 out of 5 patients (MN1 and MN2) had a normal eGFR-MDRD (≥90 mL/min/1.73 m^2^), 2 out of 5 patients (MN3 and MN5) had mild values (60–89 mL/min/1.73 m^2^), and 1 patient (MN4) had a mild to moderate value (52.8 mL/min/1.73 m^2^). MCD patients also all showed high levels of UPCR >500 mg/g, and eGFR-MDRD values ranged from normal to mild. In all LN patients, the UPCR levels were high (>500 mg/g). An LN2 patient had a low eGFR-MDRD of 12.1 mL/min/1.73 m^2^ and developed end-stage renal disease. The eGFR-MDRD values in LN1, LN3, and LN4 were normal, whereas the eGFR-MDRD value in LN5 was low (25.4 mL/min/1.73 m^2^).

### 2.2. Histopathologic Findings at Renal Biopsies of GN Patients

An IgAN3 patient was classified to class IV according to the renal histopathological classification (Figure 1A,E) [16]. In this patient, there was glomerular with mesangial hypercellularity without endocapillary hypercellularity. There were segmental sclerosis, extensive tubular atrophy, and interstitial fibrosis. There was no evidence of mesangial cells within cellular and fibrocellular crescents. Furthermore, immunofluorescence (IF) imaging showed mesangial IgA deposits (intensity 2+) and mild foot process effacement (FPE) of podocytes. Diffuse thickening of GBM was noticed in an MN4 patient (Figure 1B,F). This was accompanied by diffuse capillary wall thickening and massive IgG deposits (intensity 3+) in glomeruli. Electron microscopy (EM) showed extensive FPE of podocytes, which contained large amounts of subepithelial electron-dense deposits (EDD) mainly in the area covering the surface of the GBM in the same patient. In an MCD5 patient (Figure 1C,G), EM showed GBM with normal thickness but marked podocyte FPE. IF imaging of LN5 patients classified as class IV showed the presence and localization of immune complexes (IgG/M/A isotypes and complement C3/C1q deposits) in glomeruli (Figure 1D). In EM, the thickness of the GBM was diffused, with small subepithelial EDD seen in the GBM beneath podocytes that showed extensive FPE (no EM image available). Moreover, glomerular sclerosis was found in this patient.

### 2.3. Morphological Characteristics of Differentiated Human Podocytes

Temporal changes in conditionally immortalized human podocytes (CIHP-1) during differentiation towards a podocyte-like morphology were assessed using light microscopy (Figure 2). At a permissive temperature of 33 °C, undifferentiated cells grew typically of epithelial-like cells with a cobblestone shape, consisting of small rounded cells (Figure 2A). After thermoswitching to 37 °C over a period of 7 to 21 days, they were differentiated into arborized cells characterized by enlarged cell bodies with foot processes having both short rounded projections and long spindle-like projections similar to those previously described by Saleem et al. [17] (Figure 2B–D). We also observed that cells extended their cell processes toward neighboring cells and formed an interconnected multi-cellular network with slit-like spaces between neighboring cell processes that formed tight cell–cell contacts (Figure 2B,C). These were seen to form interdigitations between cells with slit diaphragm-like structures compatible with their in vivo appearance. In the glomerulus, podocytes can develop foot processes that form interdigitations with adjacent podocytes to form the glomerular filtration sieve or network.

During the process of development and migration, cells exhibit dynamic extension of plasma membrane protrusions called lamellipodia and filopodia. These are fundamental to cell shape and motility events to explore the chemical nature of the environment and to probe the rigidity and composition of the extracellular matrix (Figure 3A) [18]. Interestingly, numbers of lamellipodia and filopodia in CIHP-1 cells were dramatically increased significantly as differentiation progressed (Figure 3B–D). Moreover, the length and density of filopodia in CIHP-1 cells were also increased gradually and significantly (Figure 3E, *p* < 0.05).

### 2.4. Expression of Podocyte-Specific Markers

Expression levels of podocyte-specific markers, podocyte slit diaphragm proteins podocin and nephrin, podocyte foot process protein synaptopodin, and podocyte nuclear marker and transcription factor WT1 were analyzed in undifferentiated and differentiated cells, respectively. We used the F-actin cytoskeleton marker phalloidin to visualize cytoskeletal rearrangement during differentiation. In immunostaining analysis, podocin expression was present only in differentiated CIHP-1 cells (Figure 4A). Nephrin and WT1 were detected in nuclei of both undifferentiated and differentiated CIHP-1 cells (Figure 4B,C). The expression of synaptopodin and WT1 was further verified by Western blot analysis at different time points. Synaptopodin plays an important role in the maintenance of the integrity of the podocyte actin cytoskeleton. It preserves the dynamic plasticity of podocyte foot processes, thereby providing protection against proteinuria. We observed that synaptopodin expression was absent in undifferentiated CIHP-1 cells, while differentiated CIHP-1 cells showed obvious expression of this molecule (Figure 4D, left panel). WT1 was expressed in both undifferentiated and differentiated CIHP-1 cells (Figure 4D, right panel). Notably, differentiated CIHP-1 cells exhibited gradually increasing protein expression of synaptopodin and WT1 during the time course of differentiation up to day 14.

### 2.5. A Microfluidic Glomerulus-on-a-Chip of Glomerular Structural and Functional Features

We investigated whether our system could support the culture of CIHP-1 and RFP-HGMVECs separately. As shown by 3D reconstruction of Z-stack confocal images of the glomerular chip, at 8 days of co-culture, CIHP-1 (nephrin-FITC, green) and RFP-HGMVECs (red) formed intact monolayers on outer and inner surfaces of the insert membrane in the chip, respectively (Figure 5A). Moreover, average TEER of microfluidic devices under baseline conditions showed differences (between 5 and 15 Ohm∙cm^2^) in barrier integrity from 2 to 14 days (Figure 5B). In addition to CIHP-1 (epithelium) and RFP-HGMVECs (endothelium), the GFB also comprised the GBM, which could separate the two cell types. The GBM contributes to the size selectivity of the GFB and provides significant structural integrity [19,20]. Collagen type I was utilized for coating both surfaces of the membrane, which supported cell adhesion and growth on the 3D extracellular matrix (ECM) and ensured better preservation of the podocyte phenotype as indicated by more regular F-actin distribution and nephrin positivity during elongated cell processes (Figure 5C). Taken together, these results indicate that CIHP-1 and RFP-HGMVECs can be cultured in the chip while maintaining their morphology and phenotype.

One of the most important characteristics of the GFB is permselectivity, i.e., the capacity to filter molecules based on their sizes [21,22]. To evaluate the functional integrity of the filtration barrier in our chip device, we explored whether this chip also could reconstitute key kidney functions, such as the GFB, which can restrict permeability to large molecules (e.g., albumin), but can freely filter exogenous small molecules such as dextran. We measured the passage rate of 66 kDa albumin-FITC and 10 kDa dextran-rhodamine B from the top of the insert to the bottom of chamber in the chip, i.e., from the upper endothelial compartment to the lower podocyte compartment. The diffusion of both fluorescent tracers was monitored for 2 h and 6 h, respectively. It was further quantified based on fluorescence intensity. As shown in Figure 5D, albumin was retained in the endothelial compartment even after 6 h of continuous perfusion, while approximately 48.5% of dextran was filtered into the podocyte compartment at 6 h (*p* < 0.001). Thus, the human glomerulus-on-a-chip could recapitulate the normal filtration barrier of a functional glomerulus that can accurately perform differential clearance of albumin and dextran, like the GFB in vivo.

### 2.6. Establishing Models of Glomerulonephritis (GN) in a Glomerulus-on-a-Chip

Given the limited availability of in vitro models that can closely mimic human glomerular function and disease states, we explored if the glomerular chip could model a podocyte impairment and disruption of the GFB. To test this possibility, we exposed glomerular chips to serum-free media containing either a 0.5% normal serum (controls) or 0.5% serum from patients with IgAN, MN, MCD, or LN for 24 h. As shown in Figure 6A–D, cell viability declined in all patient groups compared to that in the control group. Among GN patient groups, all patients in the IgAN group showed a statistically significant decrease in cell viability of podocytes (*p* < 0.01). We also confirmed a reduction in cell viability of podocytes in the MN1 serum-induced chip as indicated through calcein-AM fluorescent live staining (Figure 6E,F; CON vs. MN1, *p* = 0.006). Furthermore, following exposure to patient serum, our results showed that patient serum led to the breakdown and disorganization of the podocyte actin cytoskeleton, which were associated with a reduction in the expression of a podocyte-specific marker WT1 in CIHP-1 cells (Figure 7). Additionally, in the albumin leakage analysis, a classic marker of podocyte injury, the permeability of albumin was increased in most chips exposed to sera from IgAN or MN patients (Figure 8A,B), while sera from MCD and LN patients did not trigger loss of permeability (Figure 8C,D). These findings are highly relevant, as increased albumin filtration can be an early sign of glomerular dysfunction. Taken together, our results indicate that the glomerular chip device can serve as an in vitro IgAN and MN model of function and injury manifestations of the GFB.

## 3. Discussion

In this study, we report the design and application of a novel glomerulus on-a-chip, fabricated to enable studying the filtration barrier function of the glomerulus. In our chip, glomerular endothelial cells and podocytes were cultured on different sides of a porous ECM-coated membrane in the insert of a microfluidic glomerular chip device, thereby successfully reconstituting the in vivo GFB structure. By arranging the 3D system with endothelial cells on the upper side and podocytes on the lower side, we developed a reliable and sensitive molecular permeability assay enabling analysis of molecular filtration in which molecules pass from the upper to the lower compartment. The chip can prevent leakage of large molecule albumin while filtering small molecule dextran in a manner similar to the GFB in vivo, thus resembling the human GFB. Moreover, in GN models induced by GN sera on a chip, cell viability and WT1 expression of podocytes were reduced compared to controls. Albumin permeability was increased in some IgAN and MN models.

In the complex filtration barrier network of the glomerulus, podocytes are the most vulnerable components and the most sensitive to exotic toxicants, resulting in filtration barrier dysfunction and nephrotoxicity. A consequence of podocyte injury is apoptosis, leading to podocyte depletion, protein leakage, and subsequent glomerular diseases. Matsusaka et al. reported propagation of podocyte damage in vivo, where primary podocyte damage can cause secondary damage to surrounding podocytes [23]. Propagation of podocyte damage is an important pathophysiology that causes a decrease in the number of podocytes per glomerulus, leading to glomerulosclerosis [24]. Therefore, the central role of podocytes in glomerular dysfunctions necessitates reliable model systems to study normal and impaired podocyte functions at cellular and subcellular levels.

In this respect, in vitro culture of human podocytes has gained specific importance. However, there are intrinsic difficulties of growing differentiated podocytes in in vitro culture. Mature podocytes are terminally differentiated cells. This is a profound limitation in cell culture since growth-arrested, differentiated cells do not proliferate. Thus, obtaining cells in sufficient quantities for basic research is extremely difficult. To circumvent this problem, undifferentiated podocyte cultures should be expanded to a certain cell density and then be differentiated by specific stimuli. Thus, we focused on human immortalized CIHP-1 cells known to present a phenotype and function very similar to those of primary podocytes. Most importantly, they can be efficiently differentiated at a large scale without using laborious protocols, as we have previously shown [13]. Our data showed that characteristic podocyte morphology, including long spindle-like projections, fine processes, short rounded projections, and interdigitations between cells, could be observed on day 7 of differentiation. Moreover, the effect on cellular differentiation was further substantiated by differences in the length and density of filopodia and lamellipodia of podocytes. This finding is in agreement with results of previous studies showing that podocytes can gain a different phenotype with huge cell bodies and expended filopodia and lamellipodia after having undergone several mitotic events and a complete differentiation process [25]. The cytoskeletal architecture of cells became more prominently visible under fluorescence microscopy, with voluminous longitudinal bundles of actin filaments in cell bodies and processes of podocytes. In addition, consistent with previous reports [17,25,26], the study of associated alterations after subjecting podocyte cultures showed that protein levels of podocyte-specific markers (podocin, nephrin, WT-1, and synaptopodin) were increased in differentiated human CIHP-1 cells. Hence, we observed that the process of maturation in podocytes in vitro was analogous to the development and maturation of podocytes in vivo. Immortalized CIHP-1 cells could provide an ideal cell resource to study podocyte epithelium by mimicking the native glomerulus employed in therapy and modeling of renal disease.

It is widely recognized that GN is the primary cause of renal damage and that podocyte injury plays a major role in GFB alterations, contributing to the progression of glomerular diseases [27,28]. Consequently, reducing podocyte damage has become a priority. Using a 3D microfluidic glomerular chip, we simulated GN disease models by adding patient serum in podocyte and endothelial compartments to characterize the effect of exposure to GN patient-derived serum on nephrotoxicity and permeability of the GFB, and observed a decrease in cell viability in all GN patients. Apoptosis assays performed on podocytes under serum treatment also reinforced observations previously confirmed by other authors [15]. Moreover, exposure to plasma from patients with MN or focal segmental glomerulosclerosis (FSGS) induced significant renal cell apoptosis, indicating impairment of cell function in a glomerulus-on-a-chip [29].

WT1 is a zinc finger transcription factor whose function is critical for normal nephrogenesis and podocyte differentiation [30,31]. In animal models, disruption of WT1 can lead to an early GBM thickening and, finally, glomerular sclerosis [31]. In humans, WT1 reduction results in a reduced filtration area and permeability [32], with proteinuria and renal scarring [33]. We also found that WT1 expression changed from a perinuclear to a more cytoplasmic pattern and reduced after exposure to patient serum.

Albumin is one of the most abundant proteins in human blood. Loss of albumin via urine is a key clinical marker for glomerular diseases. When podocytes are injured, their main function for sieving macromolecules such as albumin is no longer maintained. Previous studies have reported results of albumin leakage on a glomerulus-on-a-chip model under exposure to serum from patients with MN and podocyte injury models inducing cytotoxic drugs of puromycin aminonucleoside or adriamycin [15,34,35,36,37]. All of them showed higher levels of albumin leakage than those in normal controls. Our results also revealed that serum exposure led to enhanced intercellular albumin diffusion in some IgAN and MN patients, similar to results of previous studies [15,38]. However, there was no statistically significant difference in the permeability of albumin in MCD and LN groups compared to the control group. This might be caused by the complex etiology of these diseases. For several decades, MCD has been considered a T-cell disorder, which release a cytokine that injures podocyte foot processes and glomerular filtration barrier. Normally, cytokine release by T-cells is transient owing to the activation of Tregs that interact with T effector cells to suppress cytokine production. The induction of Tregs led to a marked reduction in proteinuria in animal models, and most patients with MCD showed decreased levels of Tregs [39,40]. Abnormal T-cell-mediated immune responses have been implicated in MCD in which podocyte effacement and proteinuria are well described [41]. T-cells have also been implicated in the pathogenesis of LN and may follow a similarly disrupted signaling pathway resulting in effacement and proteinuria. In a different model of immune complex-mediated kidney injury (MPGN), TLR4 was identified and found to be more abundant on podocytes of cryoglobulinaemic MPGN compared to the wild type [42]. TLR4 was upregulated in active disease and, once stimulated, led to the release of chemokines. The implications of this can potentially be extended into other immune complex-mediated diseases, e.g., LN. The etiology of LN involves antibody binding to intrarenal nuclear autoantigen. Anti-dsDNA antibodies play a critical role in the pathogenesis of LN through their binding to cell surface proteins of resident kidney cells, thereby triggering the downstream activation of signaling pathways and the release of mediators of inflammation and fibrosis. They may also directly cross-react with podocyte proteins like α-actinin-4 to cause injury [43]. This complexity is reflected by the paucity of major breakthroughs in the understanding of these diseases and the lack of targeted therapies. It may also be attributed to the cell source or the lack of GFB components. Mesangial cells and their matrix form the central stalk of the glomerulus. They play a critical role in maintaining glomerular structure and function by interacting closely with endothelial cells and podocytes [44]. Podocyte injury promotes mesangial cell proliferation, while mesangial cell injury causes podocyte foot process fusion and proteinuria [41]. We did not contain mesangial cells in the chip. Thus, results of some kidney diseases of mesangial cell injury originating from inflammation of the glomeruli might limit our understanding from this chip. The GFB is a complex, highly selective filter. Future work will be necessary to improve biologic components. Thus, we do not claim that we have recapitulated a complete GFB in this study. However, we have reconstituted some crucial and complex substructures of the human GFB and assessed the reproduction of GN models on a glomerular chip, which could not be addressed with conventional 2D monoculture strategies or readily investigated with animal models. Such a chip might contribute to research and therapy development in this field.

The major benefits of our perfusable glomerular chip are multiple chambers allowing for the study of uptake and transport, its reusability, and the possibility for high-throughput application. Designing a glomerulus-on-a chip consisting of two parallel cellular layers separated by a polyethylene terephthalate membrane allowed us to study two cell types separately and enabled easy separation of two cell types after co-culture. This chip was also designed from materials that enable cells to be microscopically visible, allowing for longer real-time monitoring and imaging of cell function and health. It also enables direct visualization and quantitative analysis of biological processes of the GFB in ways that have not been achieved with traditional cell culture or animal models. The specification of the removable culture insert in a flow-controlled microfluidic device allows the outflow to be collected easily for biochemical or enzymatic assays. It presents a higher user-friendliness in terms of bridging two research fields of conventional life science research and tissue-on-a-chip. Together, the versatility of three chambers could enable recapitulation of multiple in vivo microenvironments.

In addition, new chemical entity advancement through the preclinical development pipeline is extremely costly when potential candidates are triaged based on inaccurate safety data. Preclinical test systems can often be the culprit behind false-positive results based on poor in vitro to in vivo predictions. Furthermore, chip platforms represent a variety of human organ systems that can be used to screen for drug efficacy before reaching clinical trials. Mechanisms of action can also be identified and modeled appropriately. Ethics is another advantage of using this chip in early drug development. Animal models are still needed because they represent whole organisms. However, as integrated and individual organ microsystems become more easily available, cheaper, and suitable for medium- to high-throughput screening, animal testing rates in drug development may significantly decrease. In addition to helping us understand toxicity in human tissues, a glomerular chip also allows modeling of glomerular dysfunction and disease states, thereby permitting mechanistic observation of not only drug efficacy, but also disease pathology and potential therapeutic off-target effects.

Our study has a few limitations. Although the glomerular chip cultured in our fluidic flow conditions could sensitively respond to serum-mediated nephrotoxicity, it shows a bi-directional flow. However, the in vivo flow in the vascular lumen is uni-directional. It is not recirculated. Moreover, our chip in vitro model does not contain mesangial cells, an important component of normal glomeruli. Therefore, future studies need to add inserts including mesangial cells using multiple chambers. In our system, some important microenvironmental components are missing, like immune cells. Including immune cells in these perfusable glomerular chips could aid in the assessment of the immune and inflammation responses in response to biologics or viral infections. As glomerular infiltration of immune cells is an important pathogenic mechanism of immune-related glomerular diseases, the model developed in the current study might provide new insights and a clearer clinical interpretation of the pathogenesis of GN. Another limitation of this platform is that fabricating a small-diameter glomerular microvessel with a proper thickness of GBM is still challenging. Finally, given the limitation of the small sample size, a further study with a larger sample of patients is required to provide more conclusive results.

## 4. Materials and Methods

### 4.1. Patients and Samples

Among consecutive patients diagnosed with glomerulonephritis (GN) between December 2014 and April 2018 in Seoul National University Bundang Hospital who underwent kidney biopsy, 216 were recruited in this study. We enrolled 20 patients (GN group) diagnosed by renal biopsy with IgAN (n = 5), MN (n = 5), MCD (n = 5), and LN (n = 5) in this study.

All patients had a histological diagnosis confirmed by renal biopsy. Blood samples were obtained from patients at the time of the renal biopsy. To obtain serum, blood samples were left to clot at room temperature (RT) for 30 min and centrifuged at 2000× *g* for 10 min at RT. Serum samples were stored in aliquots at −80 °C until further measurements.

### 4.2. Histopathological Examination

Kidney biopsy was performed at the time of diagnosis. Biopsy specimens were formalin fixed, paraffin embedded, sliced at a thickness of 2 or 5 μm, and histologically evaluated by light microscopy, indirect immunohistochemistry, and electron microscopy (EM) following institutional guidelines. All biopsy slides were reviewed by two pathologists blinded to clinical data of patients.

### 4.3. Cell Culture and Podocytes Differentiation

CIHP-1 cells (Ximbio, London, UK, #CVCL W186) were cultured in RPMI-1640 medium (Gibco, Grand Island, NY, USA, #11875093) with 10% fetal bovine serum (Gibco, Grand Island, NY, USA, #16000-044), 1% insulin-transferrin-selenium (Gibco, Grand Island, NY, USA, #41400-045), and 1% penicillin-streptomycin (Gibco, Grand Island, NY, USA, #15140-122) at 33 °C in a 5% CO_2_ incubator. They were used within passages 5 to 15. To induce differentiation of podocytes, CIHP-1 cells were cultured at 33 °C to proliferate. They were then transferred to 37 °C to differentiate for 8 to 14 days. Differentiation was completed when cells demonstrated a morphological change at 7 to 14 days. Primary human glomerular endothelial cells (RFP-HGMVECs, ANGIO-PROTEMIE, Boston, MA, USA, #cAP-0004RFP) were cultured in endothelial basal medium (ANGIO-PROTEMIE, Boston, MA, USA, #cAP-03), to which 10% endothelial growth supplements (ANGIO-PROTEMIE, Boston, MA, USA, #cAP-04) and 1% penicillin-streptomycin were added. They were cultured on pre-coated flasks (ANGIO-PROTEMIE, Boston, MA, USA, #cAP-01) and used within passages 4 to 10. The medium was replaced every 2 to 3 days.

### 4.4. Cell Seeding

To obtain reproducible performance from the filtration flow passing through the cell layer, we used cell culture inserts with a polyethylene terephthalate porous membrane (0.4 µm pore size, Greiner BIO-ONE, Lagoas Park, Portugal, #662641). Prior to cell seeding, both sides of the insert membrane were coated with collagen type I (100 μg/mL, Corning, Tewksbury, MA, USA, #354265) for 1 h at 37 °C in a 5% CO_2_ incubator to facilitate cell adhesion. After coating, the insert was inverted. Undifferentiated CIHP-1 cells were first seeded on the bottom of the insert at a density of 1 × 10^5^ cells/mL and incubated at 37 °C in a 5% CO_2_ incubator for 2 h to complete cell attachment. The insert was used to remove the non-adherent cells and again flipped back and immediately placed in a 24-well plate (SPL, Pocheon-si, Republic of Korea, #30024) containing the CIHP-1 medium described above and incubated overnight at 37 °C in a 5% CO_2_ incubator. After the CIHP-1 layer was formed, the inside of the insert was recoated with a quick coating solution (ANGIO-PROTEOMIE, Boston, MA, USA, #cAP-01) to improve adhesion of RFP-HGMVECs. Subsequently, RFP-HGMVECs were seeded inside the insert at a density of 2 × 10^5^ cells/mL with RFP-HGMVEC medium as described above and incubated for 2 h at 37 °C in a 5% CO_2_ incubator. Afterwards, the RFP-HGMVEC medium inside the insert was replaced with a 1:1 mixture of CIHP-1 and RFP-HGMVEC medium.

### 4.5. Design and Fabrication of the Microfluidic Glomerulus-on-a-Chip Device

Figure 9 shows a schematic overview of the glomerulus-on-a-chip design for co-culture of CIHP-1 and RFP-HGMVECs. Briefly, the chip (K-bio, Osong, Republic of Korea) comprised a single layer of a polycarbonate material with three badge chambers, four microfluidic channels, and a media reservoir at each end of a 400 μm × 220 μm plate attached to the bottom with sealing foil (LightCycler^®^ 480 Sealing Foil, Roche, Pleasanton, CA, USA). The chip consisted of three cell culture chambers, four microfluidic channels, and two media reservoirs, in which CIHP-1 and RFP-HGMVECs were co-cultured on the collagen type I-coated extracellular matrix (ECM) of the insert to form the GFB. In addition, cells were cultured simultaneously with three inserts. Three samples were obtained from one chip to aid statistical processing of results. The CIHP-1 medium was allowed to flow through device reservoirs by pipetting. The insert co-cultured with CIHP-1 and RFP-HGMVECs was then placed in each of the three badge chambers. CIHP-1 and RFP-HGMVECs were co-cultured on the chip device with their respective medium for 8 days at 37 °C in a 5% CO_2_ incubator and then continuously perfused on a perfusion rocker (Organo Flow L rocker, Mimetas, Gaithersburg, MD, USA, #M1-OFPR-L) to generate a bi-directional flow. This flow was induced by setting the rocker to an angle of 7° at an interval of 8 min, generating a mean flow rate of 2.02 μL/min with a mean shear of 0.13 dyne/cm^2^ [45], which was within the physiological range (~0.2–20 dyne/cm^2^) of shear stress experienced by a normal kidney. The medium was replaced every 2 to 3 days.

### 4.6. TEER Measurement

Integrity of cellular layers was assessed by transepithelial/transendothelial electrical resistance (TEER) during fluidic culture on a chip. Resistance measurements were obtained using an epithelial volt/ohm meter (EVOM2, World Precision Instruments, Sarasota, FL, USA) equipped with handheld chopstick electrodes (STX2, World Precision Instruments, Sarasota, FL, USA). TEER was measured once every 2 days prior to media changes. Measurements were taken for inserts placed in 24-well plates at RT after stopping fluid flow. TEER values were obtained using the Ohm’s Law Method [46]. Briefly, TEER (Ohm∙cm^2^) of the cellular layer was calculated by subtracting the blank resistance of the insert membrane without cells from the resistance measured for each sample and multiplying by the surface area of the insert membrane (0.336 cm^2^).

### 4.7. FiloQuant Analysis

To analyze filopodia length and density, the FiloQuant plugin of the ImageJ software (1.54f) was utilized [47]. Manual tracking and single image tool were utilized to analyze the filopodia from maximum intensity projections created from stacks of optical sections. Single-image FiloQuant was used to detect and measure the length and number of filopodia. Filopodia density was defined as a ratio of the number of detected filopodia to cell edge length from the FiloQuant analysis.

### 4.8. Cell Viability Assay

Viability of CIHP-1 cells on a microfluidic chip was assessed using a cell counting kit-8 (CCK-8, Dojindo, Kumamoto, Japan, #CK-04) according to the manufacturer’s instructions. After exposure to GN serum, the insert on microfluidic chip was transferred onto a 24-well plate. The CCK-8 reagent was then added to the bottom of a 24-well plate and the inside of the insert. The plate was incubated for 1 h at 37 °C. Absorbance was measured at 450 nm using a microplate reader (SpectraMax iD3, Molecular Devices, San Jose, CA, USA). Data were analyzed after normalizing with the mean of the control group exposed to normal serum.

### 4.9. Calcein-AM Viability Assay

After the chip cultured for 8 days was treated with GN serum for 24 h, the chip was analyzed using calcein acetoxymethyl ester (calcein-AM, Thermo Fisher, #L3224) to determine the viability of treated cells relative to controls. To measure uptake of calcein-AM, the medium was removed and replaced with a medium containing calcein-AM (1:2000) and Hoechst 33342 (1:500, Thermo Fisher, #62249) inside the insert and on the bottom of a 24-well plate. Following incubation at RT for 30 min and washing twice with PBS, calcein-FITC images were captured by confocal microscopy (ZEISS LSM 800 Confocal Laser Scanning Microscope) with a 485/530 nm excitation/emission filter. The confocal microscopy was performed at ×50 magnification for imaging purposes.

### 4.10. Immunofluorescence Staining

Cells were fixed with 4% paraformaldehyde (PFA, Biosesang, Yongin, Republic of Korea) for 20 min at room temperature. Fixed cells were then washed with PBS three times and permeabilized using a 0.2% Triton X-100 solution (Sigma-Aldrich, Burlington, MA, USA, #T8787) for 20 min at RT. Thereafter, cells were washed with PBS and blocked with 3% bovine serum albumin (BSA, Bovogen, East Keilor, Australia, #BSAS 0.1) for 40 min at RT. Subsequently, cells were washed with PBS and incubated with primary antibodies for podocin (1:200, Bioss antibodies, Woburn, MA, USA, #bs-6597R), nephrin (1:50, Bioss antibodies, Woburn, MA, USA, #bs-10233R-FITC), and Wilms’ tumor 1 (WT1) (1:100, Bioss antibodies, Woburn, MA, USA, #bs-6983R-A647) diluted in 1% BSA in the dark overnight on a rocker at RT. They were then incubated with donkey anti-rabbit IgG Alexa fluor 488 (1:200, Abcam, Cambridge, UK, #ab150073) diluted in 1% BSA for 2 h in the dark on a rocker at RT. Actin Green 488 (2 drops per 1 mL in 1× PBS, Invitrogen, Carlsbad, CA, USA, #R37110) or Alexa Fluor™ 647 Phalloidin (1:400, Thermo Fisher, Rockford, IL, USA, #A22287) was used to stain the actin cytoskeleton by incubating for 30 min in the dark on a rocker at RT. Cell nuclei were counterstained with DAPI (1:500, Invitrogen, Carlsbad, CA, USA, #D1306) in 1× PBS for 30 min in the dark on a rocker at RT. Fluorescence images were visualized with a confocal microscope with objectives ranging from ×200 to ×400. Three-dimensional images were acquired from Z-stack modes.

### 4.11. Albumin and Dextran Permeability Assays

To evaluate barrier filtration of the chip, 0.1 mg/mL 66 kDa albumin-FITC (Sigma-Aldrich, Burlington, MA, USA, #A9771) and 0.1 mg/mL 10 kDa dextran-rhodamine B (Sigma-Aldrich, Burlington, MA, USA, #R8881) were added to the inside of the insert and incubated at 37 °C for 1 h. After 1 h, each sample was collected from the inside of the insert (endothelial compartment) and the bottom of the channel (podocyte compartment) and dispensed in a black 96-well plate. Absorbance was measured using a microplate reader with an emission/excitation wavelength of 453/488 and 540/580 nm for FITC and rhodamine B, respectively. The diffusional permeability of the chip was assessed by measuring the rate of transport of albumin or dextran from the upper endothelial to the lower podocyte compartment. Results were normalized to the average fluorescent intensity of the control and expressed as percentages.

### 4.12. Protein Extraction and Western Blotting

Proteins were extracted with SDS buffer containing a protease inhibitor cocktail (GenDEPOT, Barker, TX, USA, #P3100-020). Protein concentration was then determined using a BCA protein assay kit (iNtRON Biotechnology, Seongnam, Republic of Korea, #21071). Proteins were separated by 10% sodium dodecyl sulfate-polyacrylamide gel electrophoresis and transferred onto nitrocellulose transfer membrane (Invitrogen, Carlsbad, CA, USA, #IB23001) using an iBlot 2 gel transfer device (Invitrogen, Carlsbad, CA, USA, #IB21001). The membrane was rinsed in TBST followed by immersing in a blocking buffer (TBST with 5% BSA) for 1 h. The membrane was incubated with primary antibodies against synaptopodin (1:1000, Abcam, Cambridge, UK, #ab224491), WT1 (1:1000, Abcam, Cambridge, UK, #ab52933), or GAPDH (1:1000, Cell Signaling Technology, Danvers, MA, USA, #2118S) in TBST overnight at 4 °C. After rinsing in TBST, the membrane was incubated with secondary anti-rabbit IgG-HRP (1:20,000, GenDEPOT, Barker, TX, USA, #SA002-500) in 5% skim milk in TBST for 2 h at RT. After final washing with TBST, signal detection was performed with an ECL detection reagent (Amersham, Chalfont, UK, #RPN2232). Immunoreactive bands were detected with a ChemiDoc MP Imaging System (BIO-RAD, Hercules, CA, USA, #12003154).

### 4.13. Modeling of Glomerulonephritis (GN)

To induce the GN model, cells were added to FBS-free CIHP-1 medium in the channel of the chip and a FBS-free 1:1 mixture of CIHP-1 and RFP-HGMVEC medium inside the insert with 0.5% GN patient serum for 24 h. After 24 h, serum-supplemented media were removed from the chip. Cell viability, albumin permeability, and immunostaining assay were then performed as described earlier. Normal human serum (Millipore, Billerica, MA, USA, #S1-100ML) was used as a control.

### 4.14. Statistical Analysis

Normally distributed data are presented as the means ± SD and were analyzed using the independent *t*-test or one-way analysis of variance (ANOVA) with Tukey’s post hoc test. Non-normally distributed data are presented as the median (interquartile range) and were analyzed using the Mann–Whitney U-test or the Kruskal–Wallis test. Statistical analysis was performed with PASW Statistics version 18.0 (IBM Inc., Chicago, IL, USA). A *p* value of <0.05 was considered statistically significant.

## 5. Conclusions

We created a unique glomerulus-on-a-chip upon which the physiological structure and function of the human GFB could be reproduced. This chip can be used to detect changes in cellular morphology and protein expression. It can also be developed into a GN model by exposing it to patient-derived serum. Of pivotal relevance for the study of glomerular diseases, a glomerular chip model offers an opportunity for glomerular disease modeling, a feature that cannot be replicated using traditional in vitro models. The development of a controllable and user-friendly glomerular chip device and system could provide a potential technical platform to conduct drug screening, advance our understanding of glomerular function and mechanisms of glomerular disease pathophysiology, and facilitate the establishment of disease models to guide therapeutic discovery of glomerulopathy.

## Figures and Tables

**Figure 1 ijms-25-05121-f001:**
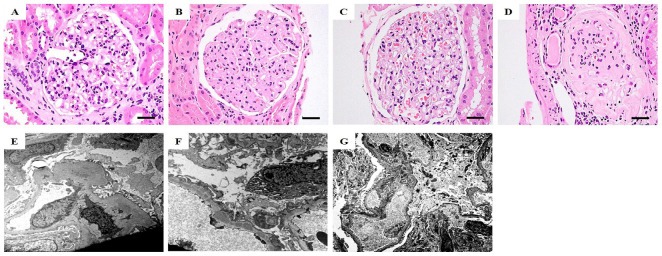
Representative histologic features via hematoxylin and eosin (H&E) staining (**A**–**D**) and electronic microscopy (**E**–**G**) of renal biopsy tissue derived from IgAN3 (**A**,**E**), MN4, (**B**,**F**), MCD5 (**C**,**G**), and LN5 (**D**). Scale bars: (**A**–**D**) 20 μm; (**E**,**G**) 5 μm; (**F**) 2 μm. Magnifications: (**A**–**D**) ×400; (**E**) ×6000; (**F**) ×8000; (**G**) ×4000.

**Figure 2 ijms-25-05121-f002:**
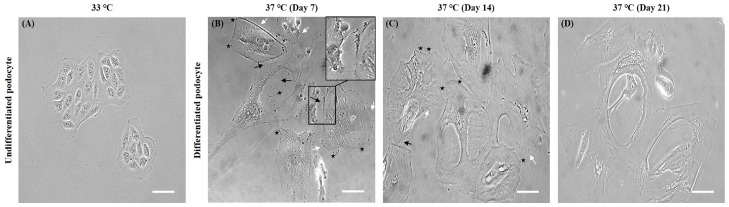
Morphology of human CIHP-1 at each stage of differentiation. Representative images demonstrating the morphology of CIHP-1 before differentiation at permissive temperature 33 °C (**A**) and after differentiation at non-permissive temperature 37 °C (**B**–**D**). Phase-contrast microscope images showing undifferentiated cobblestone-like cells without processes (**A**) and differentiated cell indicating short rounded projections (asterisk), long spindle-like projections (white arrow), and cell–cell contacts (black arrow and magnified insert) with interdigitations between cells (**B**–**D**). Scale bars: 50 μm. Magnifications: ×200.

**Figure 3 ijms-25-05121-f003:**
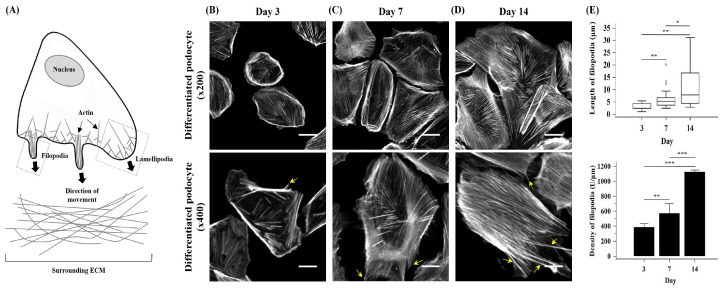
Formation and quantification of filopodia length and density of in differentiated CIHP-1 cells during the differentiation process. (**A**) Schematic diagram showing the formation of actin-rich lamellipodia and filopodia in an attempt to invade through the surrounding extracellular matrix (ECM) and migrate towards a specific direction. (**B**–**D**) Representative images of progressively extended lamellipodia and filopodia (yellow arrow) protrusions during podocyte differentiation and development. (**E**) Quantification of the length and density of filopodia (×200). Scale bars: (**B**–**D** upper) 50 μm; (**B**–**D** lower) 20 μm. Magnifications: (**B**–**D** upper) ×200; (**B**–**D** lower) ×400. Data are presented as the median (interquartile range) or mean ± standard deviation. * *p* < 0.05, ** *p* < 0.01, *** *p* < 0.001.

**Figure 4 ijms-25-05121-f004:**
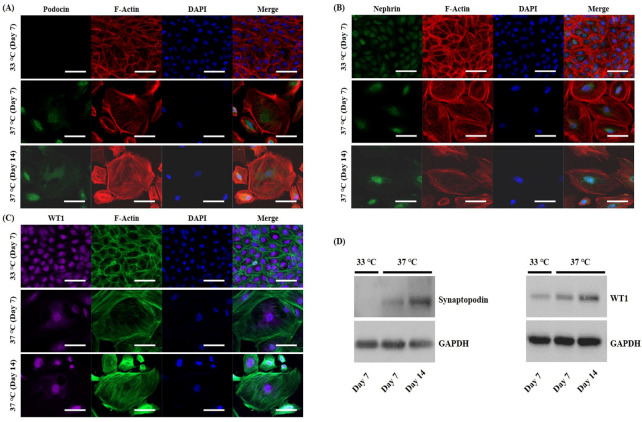
Immunofluorescence and Western blot analysis of podocyte-specific markers in undifferentiated and differentiated CIHP-1 cells. Representative immunofluorescence staining for (**A**) podocin (green), (**B**) nephrin (green), (**C**) WT1 (purple), F-actin (red or green), and nuclei (DAPI, blue) in undifferentiated and differentiated CIHP-1 cells. Scale bars: 50 μm. Magnifications: ×200. (**D**) Western blotting of synaptopodin (99 kDa, left panel) and WT1 (55 kDa, right panel) using total protein extracts from undifferentiated and differentiated CIHP-1 cells. GAPDH (37 kDa) was used as a loading control.

**Figure 5 ijms-25-05121-f005:**
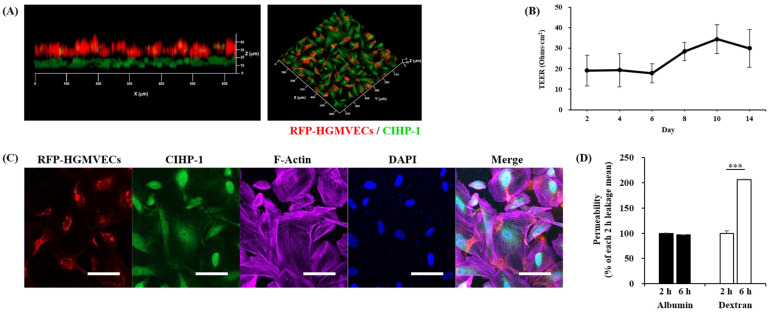
Modeling the glomerular structural and functional filtration barrier in a microfluid glomerulus-on-a-chip. (**A**) Three-dimensional visualization of Z-stack images showing the formation of glomerular endothelial cell (RFP-HGMVEC) and podocyte (CIHP-1) monolayers in the chip. (**B**) TEER measurements showing the barrier integrity of co-culture devices under baseline conditions. (**C**) Representative fluorescence images for glomerular endothelial cells (RFP-HGMVECs, red) and podocytes (CIHP-1, nephrin-FITC, green) grown on inner and outer membranes of the insert in the chip, respectively, after co-culture for 8 days. (**D**) Quantification of filtration barrier permeability to albumin-FITC (66 kDa, green) and dextran-rhodamine B (10 kDa, red) at 2 and 6 h. Data are presented as mean ± standard deviation. *** *p* < 0.001. F-actin (purple), nuclei (DAPI, blue). Scale bars: 50 μm. Magnifications: ×200.

**Figure 6 ijms-25-05121-f006:**
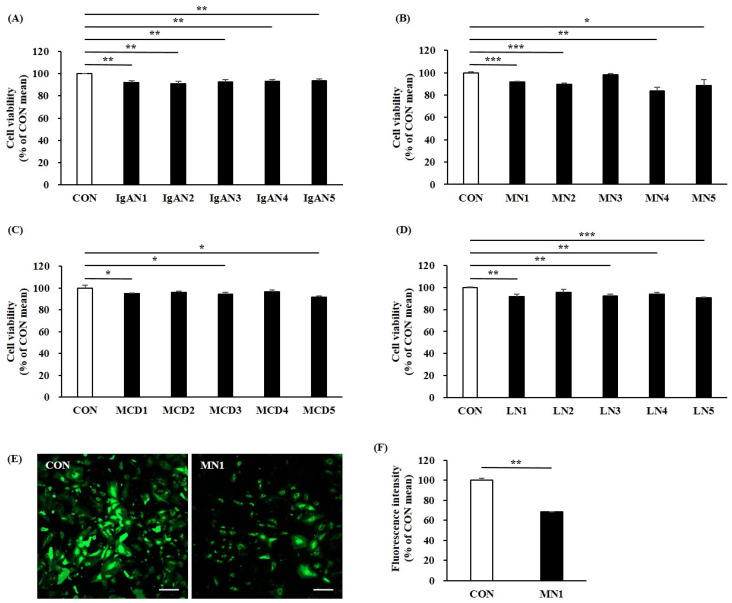
Viability of CIHP-1 cells on a chip after exposure to serum for 24 h. Quantitative assessment of cell viability of patients with IgAN (**A**), MN (**B**), MCD (**C**), or LN (**D**) using the CCK-8 assay. (**E**) Representative images of CIHP-1 cells stained with calcein-AM (green, live cell). (**F**) Quantification of fluorescence intensity of calcein-AM stained CIHP-1 cells. Data are presented as mean ± standard deviation. Scale bars: 200 μm. Magnifications: ×50. * *p* < 0.05. ** *p* < 0.01, *** *p* < 0.001.

**Figure 7 ijms-25-05121-f007:**
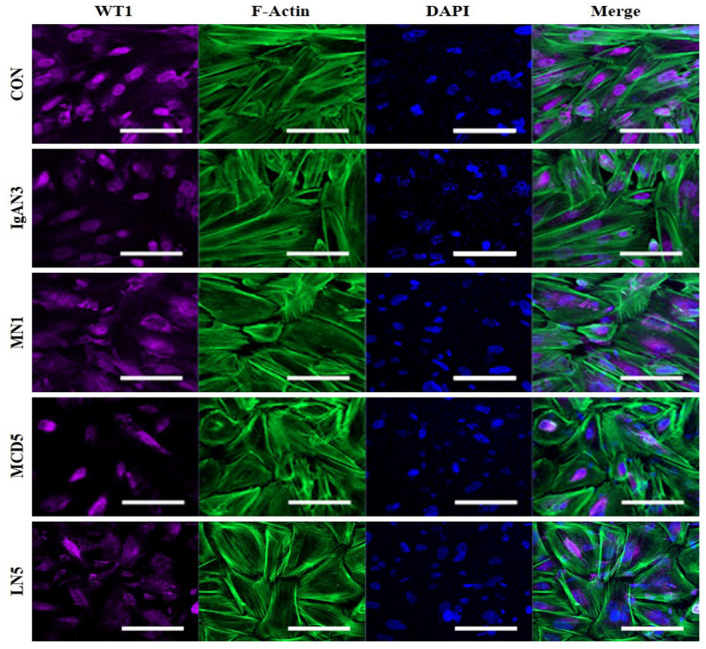
WT-1 expression in CIHP-1 cells subjected to glomerulonephritis-derived serum in a glomerular chip device. Confocal microscopic images of WT-1 staining in CIHP-1 cells after exposure to serum from patient with IgAN3, MN1, MCD5, or LN5 for 24 h. Scale bars: 50 μm. Magnifications: ×200.

**Figure 8 ijms-25-05121-f008:**
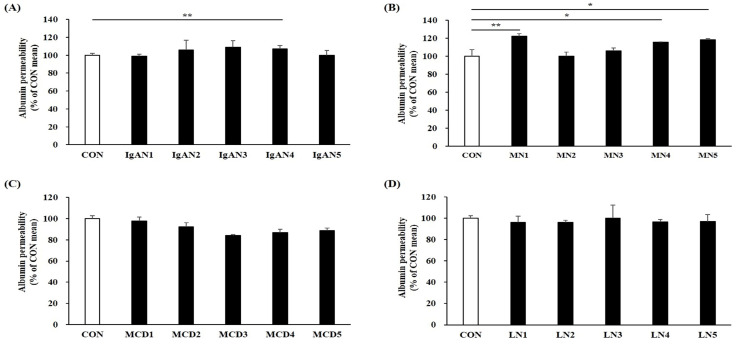
Change in the glomerular barrier permeability after exposure to GN serum in the glomerular chip system. Barrier permeability to fluorescence-labeled albumin under exposure to sera from patients with IgAN (**A**), MN (**B**), MCD (**C**), or LN (**D**). Data are presented as mean ± standard deviation. * *p* < 0.05, ** *p* < 0.01.

**Figure 9 ijms-25-05121-f009:**
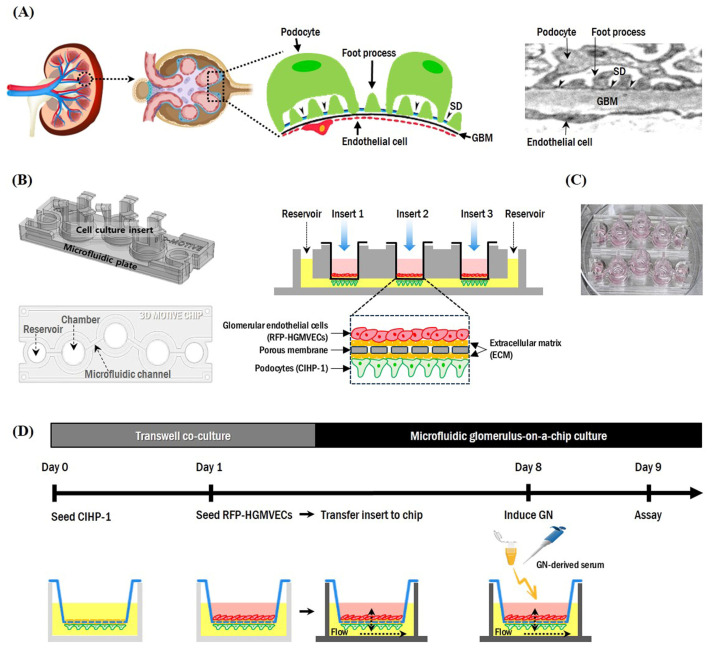
Schematic drawing of a three-dimensional microfluidic glomerulus-on-a-chip device to recapitulate the structure and function of the kidney glomerular filtration barrier. (**A**) Illustration (left panel) and an electron micrograph (right panel) of the glomerular filtration barrier (GFB). The GFB comprises podocytes with their foot processes, glomerular basement membrane, and fenestrated endothelial cells. Foot processes are interconnected by slit diaphragms (arrowheads). (**B**) Architecture of the glomerular chip. The chip consists of three cell culture chambers, four microfluidic channels, and two media reservoirs, in which podocytes (CIHP-1) and glomerular endothelial cells (RFP-HGMVECs) are co-cultured on the collagen type I-coated extracellular matrix of the insert to form the GFB. (**C**) Real image of the microfluidic chip in a cell culture dish (100 mm). (**D**) Workflow for evaluating the integrity and reproducibility of GFB and for developing a glomerulonephritis (GN) model into a microfluidic chip. CIHP-1 cells were first seeded on the bottom surface of a cell culture insert. Then, RFP-HGMVECs were seeded on the inside surface of the insert. Patient serum was added to both the chamber and the inside of the insert on day 8 to induce the GN model.

## Data Availability

Data are contained within the article and Appendix A.

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
