# Peer review of "Characterizing Glomerular Barrier Dysfunction with Patient-Derived Serum in Glomerulus-on-a-Chip Models: Unveiling New Insights into Glomerulonephritis"

_ijms, 2024, doi:10.3390/ijms25105121_

Round 1
Reviewer 1 Report
Comments and Suggestions for Authors
The authors examined the mechanism of glomerular barrier dysfunction by serum derived from patients with glomerulonephritis using a newly created Glomerulus-on-a-Chip model. This is a fascinating study and well organized. However, from a clinician's point of view, I have the following queries.
1. In this study, the authors showed that serum from IgAN and MN patients increased albumin permeability in the authors' Glomerulus-on-a-Chip model, but not in minimal change disease (MCD) or lupus nephritis (LN). Although MCD and LN are also diseases that cause severe proteinuria, these results should be discussed in more detail to determine what mechanistic differences or podocyte injury mechanisms cause proteinuria. For example, during the active phase of glomerulonephritis involving IgAN and LN, clinically significant loss of podocytes in the urine is often observed. This study discusses only the impairment of the barrier function constituted by podocytes, but what is the impact on podocyte adhesion?  Specifically, is it possible for this model to detect an effect on the adhesive properties of the basement membrane to which podocytes are attached on top in glomeruli?
2. In the same context as above, I understand that the main focus of this study is to create a Glomerulus-on-a-Chip model that is useful to study the glomerular barrier mechanism, but if the authors are going to study the mechanism of damage to the barrier mechanism in glomerulonephritis, the patient from whom serum samples were taken background should be explained in more detail. In particular, the extent to which the patients were (or were not) accompanied by urinary protein should be noted.
Author Response
Dear reviewer,
Thank you for your thoughtful and important comments.
Comments and Suggestions for Authors
The authors examined the mechanism of glomerular barrier dysfunction by serum derived from patients with glomerulonephritis using a newly created Glomerulus-on-a-Chip model. This is a fascinating study and well organized. However, from a clinician's point of view, I have the following queries.
- In this study, the authors showed that serum from IgAN and MN patients increased albumin permeability in the authors' Glomerulus-on-a-Chip model, but not in minimal change disease (MCD) or lupus nephritis (LN). Although MCD and LN are also diseases that cause severe proteinuria, these results should be discussed in more detail to determine what mechanistic differences or podocyte injury mechanisms cause proteinuria. For example, during the active phase of glomerulonephritis involving IgAN and LN, clinically significant loss of podocytes in the urine is often observed. This study discusses only the impairment of the barrier function constituted by podocytes, but what is the impact on podocyte adhesion? Specifically, is it possible for this model to detect an effect on the adhesive properties of the basement membrane to which podocytes are attached on top in glomeruli?
Reply: As per your comment, the pathogenesis of MCD and LN was additionally described in the discussion section to determine what mechanistic differences or podocyte injury mechanisms cause proteinuria (page 10, line 377-395). Our model can detect effects on the adhesive properties of the basement membrane to which podocytes are attached on top in glomeruli. The strength and integrity of cellular layers attached the ECM can be assessed by TEER measurements during fluidic culture on a chip. As shown in Figure 1, the degree of adhesion of podocytes attached to the bottom of the cell culture insert membrane on a chip can be confirmed through a microscope using a 3D analysis of z-stack confocal images. In our chip, it can be seen that the attached podocytes (nephrin-FITC, green) and endothelial cells (red) form intact monolayers on outer and inner surfaces of the insert membrane in the chip, respectively (Figure 1A). After exposure to patient serum, the attached podocytes detach from the membrane (yellow arrows) (Figure 1B). The expression of nephrin, a podocyte marker, is also decreased. The number of podocytes attached on a microfluidic chip was quantified using a cell counting kit-8 assay to assess cell viability. Additionally, we measured the amount of albumin released out of the insert by podocyte death and podocyte detachment lead to loss of podocyte-podocyte tight junctions. In these experiments, the attachment of podocytes to the ECM can be detected.
- In the same context as above, I understand that the main focus of this study is to create a Glomerulus-on-a-Chip model that is useful to study the glomerular barrier mechanism, but if the authors are going to study the mechanism of damage to the barrier mechanism in glomerulonephritis, the patient from whom serum samples were taken background should be explained in more detail. In particular, the extent to which the patients were (or were not) accompanied by urinary protein should be noted.
Reply: As per your comment, the uPCR (urine protein/creatinine ratio) and eGFR-MDRD (estimated glomerular filtration rate-modification of diet in renal disease), which were used to identify proteins in urine, were additionally described in the results section (page 3, line 99-112).

Reviewer 2 Report
Comments and Suggestions for Authors
This is a very interesting manuscript that aims to respond to a current demand, establishing a new in vitro method to evaluate glomerular damage through the creation of an organ on a chip. The manuscript is correct and could be published as is. I have some suggestions/doubts that have arisen from reading it
Although the limitations are reflected in the text, I think there are some more that should be mentioned or explained. I believe that this model could be used to understand the cellular alterations associated with different pathologies or exposure to toxins, but I believe that with this system it is very difficult to evaluate the true glomerular filtration, that is, the real functionality of the glomerulus. Is there any strategy to estimate if the filtration is being carried out correctly with these systems?
Another limitation is that external stimuli to the kidney that could affect filtration are not taken into account, such as the interaction with the immune system or with different hormones. Do you think this could affect the results and their correct clinical interpretation?
Author Response
Dear reviewer,
Thank you for your thoughtful and important comments.
Comments and Suggestions for Authors
This is a very interesting manuscript that aims to respond to a current demand, establishing a new in vitro method to evaluate glomerular damage through the creation of an organ on a chip. The manuscript is correct and could be published as is. I have some suggestions/doubts that have arisen from reading it
- Although the limitations are reflected in the text, I think there are some more that should be mentioned or explained. I believe that this model could be used to understand the cellular alterations associated with different pathologies or exposure to toxins, but I believe that with this system it is very difficult to evaluate the true glomerular filtration, that is, the real functionality of the glomerulus. Is there any strategy to estimate if the filtration is being carried out correctly with these systems?
Reply: The chip simulated some of the important cellular structures of the glomerular filtration layer. Changes in permeability were accompanied by alternations in cell morphology, including opening of cell-to-cell tight junctions and loss/damage of cells. There are glomerular component and structural limitations to fully assessing the real filtration function of the glomeruli in our system. In future studies, we will analyze and quantify albumin-FITC uptake in podocytes as well as changes in albumin filtration rate to recapitulate a more complete glomerular filtration barrier by adding glomerular constituent cells and vascularization. Moreover, chips are designed from materials that enable cells to be microscopically visible, further studies will plan to monitor and image albumin permeability rate by exposure to toxic substances using real-time cell imaging system (JuLI™ Stage).
- Another limitation is that external stimuli to the kidney that could affect filtration are not taken into account, such as the interaction with the immune system or with different hormones. Do you think this could affect the results and their correct clinical interpretation?
Reply: The challenge in this field is to recapitulate in vivo physiology of at least a subset of functions that have value to the research, clinical or drug development communities, and then to progressively add additional functions over time. The development of tissue/organ-on-a-chip models that can present an optimal cell culture for simulating an in vivo-like microenvironment including hormonal stimulation, the immune system, the lymph, the microbiome, and vascularization, is a key challenge not only for us but also for researchers in this field. In our system, important microenvironmental components didn’t contain, like the immune system and hormone. These limitations have been added in the discussion section (page 11, line 444-450). Thus, we do not claim to have rebuilt a complete glomerulus in this study. As per your comment, I think that adding circulating immune cells (T cells, B cells, macrophages, neutrophils), non-immune cells (fibrocytes), and hormones into the chip in further studies might provide new insights and a correct clinical interpretation in the pathogenesis of GN. However, although the use of various kinds of attaching cells can be easy, culturing suspending cells remains a major challenge in tissue/organ-on-a-chip development. Hence, the optimization of a new cell culture system is challenging and more studies will be necessary to improve upon the biologic components to create the in vivo–like microenvironment.